# PIXEL REDRAWN FOR A ROBUST ADVERSARIAL DEFENSE

## ABSTRACT

Recently, an adversarial example becomes a serious problem to be aware of because it can fool trained neural networks easily. To prevent the issue, many researchers have proposed several defense techniques such as adversarial training, input transformation, stochastic activation pruning, etc. In this paper, we propose a novel defense technique, Pixel Redrawn (PR) method, which redraws every pixel of training images to convert them into distorted images. The motivation for our PR method is from the observation that the adversarial attacks have redrawn some pixels of the original image with the known parameters of the trained neural network. Mimicking these attacks, our PR method redraws the image without any knowledge of the trained neural network. This method can be similar to the adversarial training method but our PR method can be used to prevent future attacks. Experimental results on several benchmark datasets indicate our PR method not only relieves the over-fitting issue when we train neural networks with a large number of epochs, but it also boosts the robustness of the neural network.

## 1 INTRODUCTION

Deep neural networks have exhibited very accurate results, but, are also vulnerable to adversarial examples (Szegedy et al., 2013). Adversarial example is a generated output that induces a trained machine to misclassify a test datum with a high probability. In other words, the adversarial example can fool the trained machine to fail its task (Dhillon et al., 2018b; Goodfellow et al., 2014; Szegedy et al., 2013). We call the method that generates adversarial examples as an attack technique. For instance, in image classification, an attack technique is creating an adversarial image by perturbing some pixels of the original image in which the adversarial image is remained to be perceptible by the human. Obviously, the lesser the pixels to be perturbed, the better the attack technique it is. Defense techniques to prevent those attacks usually need to produce a robust neural network model which can detect adversarial examples or label the adversarial examples correctly. There are many researchers who have studied both attacking and defense techniques, and it is generally believed that the defense techniques are more challenging than the attack techniques.

In this paper, we focus on the defense technique that generates a robust neural network. For this purpose, we have to understand the fundamental idea of generating the adversarial example. In image classification, usually, an adversarial image is generated by perturbing its pixels' value (for example, $L_2$ or $L_\infty$ attack). If we regard the perturbed pixels to be formed by adding some noises, then the technique of creating the noises can be recognized as a dropout (Srivastava et al., 2014) (if the perturbed pixel value is zero) or other regularization techniques (Adeli & Wu, 1998; Krogh & Hertz, 1992; Nowlan & Hinton, 1992). The neural network with the regularization method usually shows higher performance than the one without the regularization method (normal testing). From another viewpoint, we can regard adversarial examples as a subset from the training dataset universe. Due to the difficulty of collecting all possible examples in the world, we usually just use the clean (unperturbed) examples (Kurakin et al., 2016b; Yuan et al., 2017) as the representative of our training dataset.

In order to solve the problem, we propose Pixel Redrawn (PR) method, which is a pre-processing method that redraws a pixel value of the original image into a different pixel value. The PR method is motivated by two observations. The first observation is based on the lesson provided by Carlini and Wagner in (Carlini & Wagner, 2017), which the randomization can increase the distortion required

for attack. The paper has mentioned that the most effective defense technique so far is the dropout randomization which has increased nearly five times more difficulty in generating the adversarial examples on CIFAR dataset. We agree that the randomization method makes it difficult to compute the derivation during the back-propagation and it could replace the perturbed pixel with another value by some random chances during the feed-forward.

The second observation is that humans usually can recognize images regardless of the colors in which the image has been drawn (e.g. in various colors or in gray level). This indicates that we usually (human) may not need to have a colorful image just for image classification. We discuss this in more detail with our proposed method in Section 2.

In this study, we focus on the following:

- We analyze the influence of our PR method toward a normal deep neural network without any adversarial machine learning.

- We evaluate the effectiveness of our PR method applied in the deep neural network against the adversarial example.

- And, we compare our PR method with a random noise injection.

Our research contributions can be summarized as follows:

- Our PR method effectively generates adversarial training images that are not covered in the original dataset.

- Hence, our PR method increases the robustness in neural network training.

To promote reproducible research, we release the implementation of our defense. [1]

## 2 PIXEL REDRAWN METHOD

### 2.1 PRELIMINARIES

Let $X$ be the image for the input of neural networks. If the size of the image is $m \times n$, then $\{x_i | x_i \in X\}$, where $x_i = x_1, x_2, \ldots, x_{m \times n}$ is the pixel of the image. A classifier $F(\cdot)$ is a function to produce a predicted label $\widehat{Y}$ for $X$. Let $Y$ be the true label of image $X$.

An attack technique $A(\cdot)$ is a function to generate an adversarial image $X^{adv}$ from $X$ with (or without) the knowledge of $F(\cdot)$.

### 2.2 OUR APPROACH

As aforementioned, PR is a pre-processing method $PR(\cdot)$ which reproduces a new image by redrawing the pixels of the original image. We called the generated image as PR image $X^{pr}$. We list the three main steps of PR method as follows:

1. **Discretization** Let $C$ be the range of color $C \in [0, 255]$. Note that we use a normalized pixel value $C \in [0, 1]$ in the experiment section (Section 4). If we discretize the color into $k$ sets, then $\{c_i | c_i \in C\}$, where $c_i = c_1, c_2, \ldots, c_k$ is the sub-range of color in $C$. For each $c_i$, the range starts from a minimum value $s^{pr}_{i_{min}}$ and ends with a maximum value $s^{pr}_{i_{max}}$ (e.g. $c_1 = [s^{pr}_{1_{min}}, s^{pr}_{1_{max}}] = [0, 51]$ if $k = 5$ and $C$ is equally divided).

2. **Prediction** We train a PR model (describe more in Section 2.3) and then we use it to predict each pixel of an image.

$$\hat{y}^{pr}_i = wx_i + b \tag{1}$$

where $\hat{y}^{pr}_i$ is a prediction label of a pixel from PR model, $w$ is a weight of PR model, $x_i$ is a pixel of an image, and $b$ is a bias of PR model (shown in 'PR process' of Fig. 1).

---

[1] We modify and extend the scripts from CleverHans. Our scripts are available at https://github.com/canboy123/pixel-redrawn

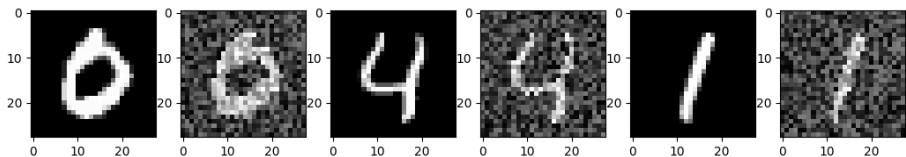

Figure 1: Pixel Redrawn concept with MNIST example.

Figure 2: Example of original images and PR images from MNIST dataset. From a pair of an image, the left-hand side is the original image and the right-hand side is the PR image.

3. **Mapping** We redraw the pixel with the sub-range of color $c_i$ that has predicted from PR model.

$$c_i = colorset(y_i^{pr}) \tag{2}$$

$$x_i^{pr} = random_{c_i}(s_{i_{min}}^{pr}, s_{i_{max}}^{pr}) \tag{3}$$

where $colorset(\cdot)$ is the color set that has defined by the user, $c_i$ is the sub-range of color with a minimum value $s_{i_{min}}^{pr}$ and a maximum value $s_{i_{max}}^{pr}$, and $random_{c_i}(\cdot)$ is a random function that generates a random value between $s_{i_{min}}^{pr}$ and $s_{i_{max}}^{pr}$.

In Fig. 1, we perform pre-processing (PR process) for the original image. We reassign each pixel of the image to a random value, such that the range of the random value is based on the chosen sub-range of color which is predicted from PR model. The newly generated PR image is then fed into a neural network.

We depict some examples from a real dataset (i.e. MNIST (LeCun et al., 1998) dataset) with PR method in Fig. 2. The intuition of redrawing each pixel with a certain value in an accepted interval is that, for a certain training image, the training dataset does not include all possible combination of the training image with human-acceptably varying pixel values. With the PR method, we try to include those missing data in the training phase. Another advantage of using the PR method is that we can generate more accurate neural network models (discussed in more detail in Section 2.5). In the following sub-sections, we explain PR model for effectively generating new pixel redrawn images by our PR method.

## 2.3 GENERATING PIXEL REDRAWN MODEL

Given a pixel value from the original training image $x_i$, PR model predicts the output pixel value $\hat{y}_i^{pr}$, perturbed within an accepted range $c_i$, for the corresponding pixel location of the new image $x_i^{pr}$. Since we redraw each pixel of the image (either grayscale or color) independently which the pixel is the only input for the PR model, and we need the PR model to produce several outputs, then single-layer Perceptron is qualified in this case. Firstly, we train the PR model by generating a batch of random examples with the corresponding labels initialized with the $c_i$. After the PR model is converged, we predict each pixel of the original image with the newly trained PR model. Lastly, we assign a predicted output value of the winning node at the last layer in the trained PR model to the corresponding pixel location of the new image. Note that the output value is within the accepted range which is initialized by the user. The final output should look similar as the right-hand side image of a pair image in Fig. 2).

## 2.4 Pseudo-code of the Pixel Redrawn Method

In order to have a clear description of our proposed method, we provide the pseudo-code of the PR method in Algorithm 1. First, we initialize the settings of the PR models, which include the range of each color value $c_i$, the number of the input (always 1) and output ($k$), the number of the epochs/iterations, and some random weights. Then, we train the PR model by generating some random inputs with a corresponding label which the label is based on the range of each color value. After we train the PR model, we generate a PR image ($X^{pr}$) by using the trained PR model to redraw all pixels of the original image ($X$). Next, we use the newly generated PR image ($X^{pr}$) as the input of the neural network during the training phase and testing phase. In the end, the trained network show the output label of the test data.

---

**Algorithm 1** The pseudo-code of the Pixel Redrawn Method

---

1: **Input:** Image dataset.
2: Initialize $k$, $c_i$ where $c_i = c_1, \ldots, c_k$, random weights
3: **Preprocessing phase:**
4: Train a PR model with a batch of random generated values which has assigned with a proper label
5: Predict each pixel of the original image (training and test data) with the trained PR model
6: Generate a PR image $X^{pr}$ by assigning a new pixel value for each pixel of the original image which is based on the output of the trained PR model
7: **Training phase:**
8: Train a neural network with $X^{pr}$
9: **Testing phase:**
10: Classify $X^{pr}$ with the trained neural network
11: **Output:** The class label of the test image

---

## 2.5 Extension of the Pixel Redrawn Model

One direct extension is using multiple PR models. We train $d$ PR models and predict each pixel of the original image with one of the trained PR models. In other words, we randomly select one PR model out of $d$ PR models to predict a pixel of the original image. Then, we repeat the same steps until all pixels are redrawn.

Another extension is partially converged PR models (less accurate) to redraw a pixel to a totally different color value. The partially converged PR model is generated when we prematurely stop the training phase on purpose after a few epochs so that the model is not fully trained. For example, if a black and white ($k = 2$) PR model has 70% accuracy to predict a pixel either black or white color, then a black color pixel can have 30% chance to be converted into a white color. Note that this may or may not influence the content of the image depending on the number of the pixels that are different from the original input. In the extreme case, PR method may able to change an eye color from black to blue for example, by using the partially converged PR models. The motivation of using the partially converged PR models is because the adversarial example changes a pixel value into a different value which the changed value is far from the original value ($L_\infty$ attack).

In summary, using several PR models (including some partially converged PR models) increase the robustness of the neural network.

## 3 Experimental Settings

### 3.1 Dataset

For experimental analysis of our propose methods, we use three public benchmark datasets in this study. The datasets include MNIST, Fashion MNIST (Xiao et al., 2017) and CIFAR-10 (Krizhevsky et al., 2010). For the CIFAR-10 dataset, we generate another grayscale CIFAR-10 for the purpose of analyzing the influence of our proposed method on the color image. MNIST and Fashion MNIST have 60,000 training images and 10,000 test images associated with a label from ten classes. The size of each image is 28×28 grayscale. CIFAR-10, however, has 50,000 training images and 10,000 test images with ten classes. Each image is 32×32 color image.

We use a basic convolutional neural network (CNN) (Krizhevsky et al., 2012) as the neural network architecture for all datasets. The neural network architecture is shown in Fig. 1. We use three convolutional layers and one fully connected layer. The convolutional layers are followed by the Rectifier Linear Unit (ReLU) activation function. The fully connected layer is followed by the Softmax activation function.

## 3.2 ATTACK TECHNIQUE

There are two kinds of attacks, which are white-box attack and black-box attack. In the white-box attack, attackers know all parameters of the attacked model, whereas in the black-box attack, attackers have no knowledge about the parameters of the attacked model. In this experiment, we run the white-box attack because it is the hardest to defend. The attack can be targeted attack or untargeted attack. In the targeted attack, the attacker tries to deceive the trained network to misclassify the datum as the targeted label. The untargeted attack, on the other hand, is fooling the trained model to misclassify the datum as any label except the true label. We choose the untargeted attack in the experiment because it is easier for the attacker than the targeted attack.

We use several state-of-the-art attack techniques for the evaluation of our proposed method. The attack techniques include Fast Gradient Sign Method (FGSM) (Goodfellow et al., 2014), basic iterative method (BIM) (Kurakin et al., 2016a), momentum iterative method (MIM) (Dong et al., 2018) and $L_2$-Carlini & Wagner's ($L_2$-CW) (Carlini & Wagner, 2016) attack. During the experiments, when we apply with the FGSM, BIM, and MIM attacks, we set $\epsilon = 0.3$ for the MNIST dataset and $\epsilon = \frac{8}{256}$ for the Fashion MNIST and CIFAR-10 datasets . For CW attack, we set 1,000 in the number of iterations to run the attack.

**FGSM** It is a fast and simple attack technique to generate the adversarial example which has proposed by Goodfellow et al. (2014). We set this technique as the baseline of attack techniques. The equation of FGSM is computed by

$$X^{adv} = X + \epsilon \, sign(\nabla_X L(X, y)) \tag{4}$$

where $\epsilon$ is the maximum perturbation allowed for each pixel and $L(X, y)$ is a loss function.

**BIM** It is an extension of FGSM by applying multiple iterations with small step size in order to obtain the least perturbations of the image. This technique is proposed by Kurakin et al. (2016a). The computation of the technique is shown in Eq. 5

$$X_0^{adv} = X, \; X_{N+1}^{adv} = Clip_{X,\epsilon}\{X_N^{adv} + \alpha \, sign(\nabla_X L(X_N^{adv}, y))\} \tag{5}$$

where $Clip_{X,\epsilon}$ is a function to clip the output image to be within the $\epsilon$-ball of X.

**MIM** It is more advanced than BIM with momentum algorithm. This technique is proposed by Dong et al. (2018). The computation of the technique is shown in Eq. 6

$$X_{t+1}^{adv} = X_t^{adv} + \alpha \cdot \frac{g_{t+1}}{||g_{t+1}||_2} \tag{6}$$

where $g_{t+1}$ is shown in Eq. 7.

$$g_{t+1} = \mu \cdot g_t + \frac{J(x_t^{adv}, y)}{||\nabla_x J(x_t^{adv}, y)||_1} \tag{7}$$

**CW** It is an efficient attack technique in finding the adversarial example with the smallest perturbations. The equation is shown as follow.

$$minimize \parallel \frac{1}{2}(tanh(w) + 1) - x \parallel_2^2 + c \cdot f(\frac{1}{2}(tanh(w) + 1)) \tag{8}$$

## 3.3 DEFENSE TECHNIQUE

We categorize the defense techniques into two different categories. They are white-box defense and black-box defense. In the white-box defense, defenders use only the known attack techniques to generate adversarial examples to be included in a training dataset (e.g. adversarial training (Goodfellow et al., 2014; Madry et al., 2017; Tramèr et al., 2017)). In the black-box defense, on the other

hand, defenders have no knowledge about the attack techniques and they try to generate a robust neural network model (e.g. stochastic activation pruning (SAP) (Dhillon et al., 2018a), input transformations (Guo et al., 2018), etc.). The white-box defense is usually performed better than the black-box defense because it can use the state-of-the-art attack technique to create adversarial examples for the training purpose and then the trained model can prevent the attacks with the similar level (or weaker levels) of the known attack technique. However, unlike the black-box defense, the white-box defense might not be able to defend strong attacks devised in the future. Due to the disability of the white-box defense in defending future attacks, the black-box defense is more reliable to be studied. Therefore, most state-of-the-art defense techniques are the black-box defense.

## 3.4 CASE STUDY WITH DIFFERENT ATTACK SCENARIOS

In this paper, we evaluate our method in several cases with different attack scenarios. The case studies are included as follows:

- **Normal:** No attack technique is used. Test the neural networks with a legitimate datum.
  $\widehat{Y} = F(PR(X))$

- **Case A:** The attackers have no knowledge of the PR method but they knows the parameters of the trained neural networks. The attackers create an adversarial image from an image with the parameters of the trained neural networks. The defense mechanism receives the adversarial image as the input for the PR model and then generate PR image for the input of the trained neural networks.
  $\widehat{Y} = F(PR(A(F, X)))$

- **Case B:** No PR method is used during the testing phase. The attackers have no knowledge of PR method but they knows the parameters of the trained neural networks. The attackers generate an adversarial image from an image with the parameters of the trained neural networks. The trained neural networks use the adversarial image as the input without being pre-processed by the PR method.
  $\widehat{Y} = F(A(F, X))$

- **Case C:** The attackers know both the PR method and the trained neural networks. The attackers produce an adversarial image from an image with the parameters of the PR method and trained neural network. The trained neural networks use the adversarial image as the input without being pre-processed by the PR method.
  $\widehat{Y} = F(A(F, PR, X))$

- **Case D:** The attackers know both the PR method and the trained neural networks. The attackers produce an adversarial image from an image with the parameters of the PR method and trained neural network. The defense mechanism receives the adversarial image as the input for the PR model and the newly created PR image is used as the input of the trained neural networks.
  $\widehat{Y} = F(PR(A(F, PR, X)))$

We illustrate some adversarial images generated from the MNIST dataset for each case in Fig. 3. Note that the adversarial example generated from the case C and D can cause high $L_0$, $L_2$ and $L_\infty$ distances because the attacker has used a distorted image instead of a clean image. In other words, the perturbed pixels from the adversarial image can be perceptible by the human.

## 4 EXPERIMENTAL RESULTS

In this study, we perform several experiments based on the following questions:

1. What is the influence of applying the PR method in different phases with a legitimate test datum?
2. What is the influence of using the PR method in different case studies?
3. What is the difference between multiple PR models and a single PR model applying in the neural networks?
4. What is the difference between the PR method and the random noise injection?

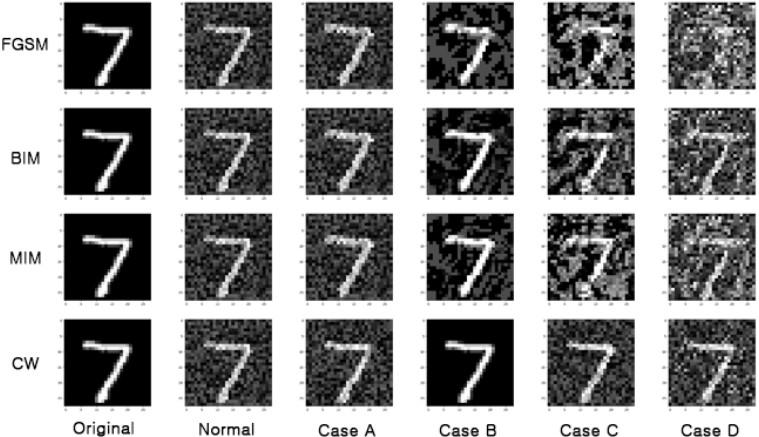

Figure 3: The example of the input images for the classifier from MNIST dataset with FGSM ($\epsilon = 0.3$), BIM ($\epsilon = 0.3$), MIM ($\epsilon = 0.3$), and $L_2$-CW attack techniques in five cases.

## 4.1 EXPERIMENT TO ANSWER QUESTION # 1

The objective of performing these experiments is to demonstrate that PR augmented training method has more advantages than common training methods. From Table 1, we produce four different results by applying the PR method in different phases. These include (1) applying no PR method, denoted as non-PR (i.e. common training method), (2) applying PR method during the training phase, (3) testing phase (4) and both phases. We analyze that applying PR method (either during the training phase or both phases) is not significantly different with non-PR case in terms of the accuracy. This means that neural networks trained with PR method show comparable performance to those trained without PR method. From the results of "Testing Phase" in Table 1, it can be seen that neural networks trained without PR (i.e. non-PR) are not sufficiently robust when they are exposed to images perturbed by PR method. In other words, we can use PR method as an attack technique to fool the trained neural networks. The accuracy for the MNIST and Fashion MNIST datasets are greatly decreased in Table 1 could be caused by over-fitting when a large number of epochs have passed. However, our method can relieve the over-fitting issue as shown in Table 1 when we apply PR method in the training phase only. The neural networks can classify clean images correctly after we train them with PR method. We conjecture PR method has performed regularization while training the neural networks. Common regularization methods (e.g. dropout, weight decay, etc.) usually are related with neural networks architecture, but our PR method is applied to the data themselves.

Table 1: Comparison results of applying pixel redrawn method during various phases.

| Dataset | Epoch | Accuracy | | | |
|---------|-------|----------|---------------|--------------|-------------|
|         |       | Non-PR | Training Phase | Testing Phase | Both Phases |
| MNIST | 50 | 0.9921 | 0.9905 | 0.8097 | 0.9893 |
|  | 500 | 0.9885 | 0.9824 | 0.4964 | 0.9879 |
|  | 1000 | 0.9893 | 0.9671 | 0.2765 | 0.9865 |
| Fashion MNIST | 50 | 0.9106 | 0.8879 | 0.3538 | 0.8724 |
|  | 500 | 0.9058 | 0.8795 | 0.3018 | 0.8670 |
|  | 1000 | 0.8964 | 0.8788 | 0.2550 | 0.8632 |
| CIFAR-10 | 50 | 0.6179 | 0.5703 | 0.5144 | 0.5386 |
|  | 500 | 0.5989 | 0.5557 | 0.4864 | 0.5323 |
|  | 1000 | 0.5873 | 0.5495 | 0.4739 | 0.5204 |
| CIFAR-10 (grayscale) | 50 | 0.5697 | 0.5099 | 0.3764 | 0.4845 |
|  | 500 | 0.5466 | 0.4864 | 0.3429 | 0.4609 |
|  | 1000 | 0.5369 | 0.4761 | 0.3370 | 0.4569 |

## 4.2 EXPERIMENT TO ANSWER QUESTION # 2

The goal of operating these experiments is to validate whether PR method can be effectively used as a defense technique against the state-of-the-art attack techniques. We conduct five cases that we have discussed in Section 3.4 in this sub-section. From Table 2, PR method can defend most attacks in the case A and D. One interesting observation that we have found in the case B is that the neural networks which have been trained with PR method during the training phase are robust to defend most attacks (except the $L_2$-CW attack) even if we do not apply PR method during the testing phase. This shows that our PR method boosts the robustness of the neural networks. For the case C, the performance of the neural networks is significantly degraded due to large increase of $L_p$ distance (i.e. the distance between the original image and the adversarial image). As we know, the case C is not practical because the $L_p$ distance has to be as low as possible in order to create an adversarial example. Although the case C is impractical, we still perform the case D to validate our PR method can increase the robustness of the neural network (except MNIST dataset with FGSM technique because of high distortion of the generated adversarial image as shown in Fig. 3). We conjecture our method which can defend the attacks effectively because it has eliminated most adversarial pixels during the pre-processing phase. Furthermore, we can see that our PR method performs better in the grayscale dataset from the comparison between CIFAR-10 and CIFAR-10 (grayscale) dataset.

Table 2: Results for several datasets in the case A, B, C, and D with 1,000 epochs and $k$=3. Note that in the case A, we classify as $F(PR(A(F, X)))$. In the case B, we classify as $F(A(F, X))$. In the case C, we classify as $F(A(F, PR, X))$. In the case D, we classify as $F(PR(A(F, PR, X)))$.

| Dataset | Attack | Accuracy, $k$=3 | | | | |
|---|---|---|---|---|---|---|
| | | Normal | Case A | Case B | Case C | Case D |
| MNIST | FGSM, $\epsilon = 0.300$ | 0.9870 | 0.9689 | 0.8787 | 0.6396 | 0.3664 |
| | BIM, $\epsilon = 0.300$ | | 0.9831 | 0.7930 | 0.5115 | 0.7697 |
| | MIM, $\epsilon = 0.300$ | | 0.9680 | 0.7504 | 0.4264 | 0.5209 |
| | $L_2$-CW | | 0.9853 | 0.6621 | 0.1526 | 0.9690 |
| Fashion MNIST | FGSM, $\epsilon = 0.031$ | 0.8632 | 0.8426 | 0.7642 | 0.6887 | 0.7430 |
| | BIM, $\epsilon = 0.031$ | | 0.8420 | 0.7271 | 0.6277 | 0.7376 |
| | MIM, $\epsilon = 0.031$ | | 0.8340 | 0.6815 | 0.5847 | 0.7137 |
| | $L_2$-CW | | 0.8450 | 0.1048 | 0.0932 | 0.8159 |
| CIFAR-10 | FGSM, $\epsilon = 0.031$ | 0.5204 | 0.3343 | 0.2265 | 0.2177 | 0.2236 |
| | BIM, $\epsilon = 0.031$ | | 0.3554 | 0.2104 | 0.2054 | 0.2216 |
| | MIM, $\epsilon = 0.031$ | | 0.3025 | 0.1904 | 0.1913 | 0.1938 |
| | $L_2$-CW | | 0.4667 | 0.1728 | 0.1654 | 0.4156 |
| CIFAR-10 (grayscale) | FGSM, $\epsilon = 0.031$ | 0.4569 | 0.3371 | 0.2217 | 0.2016 | 0.2502 |
| | BIM, $\epsilon = 0.031$ | | 0.3487 | 0.2079 | 0.1969 | 0.2633 |
| | MIM, $\epsilon = 0.031$ | | 0.3216 | 0.1888 | 0.1854 | 0.2291 |
| | $L_2$-CW | | 0.4290 | 0.1845 | 0.1727 | 0.3868 |

In addition, we compare the performance between a neural network trained with PR method (during the training phase) and a neural network trained without PR method under four different attacks, shown in Table 3. Note that we do not apply PR method during testing phase for the neural network that has been trained with PR method (case B). From Table 3, neural networks trained with PR method show higher performance than non-PR neural networks except in the normal case. This result indicates our PR method can increase the robustness of the neural network.

## 4.3 EXPERIMENT TO ANSWER QUESTION # 3

The target of implementing these experiments is to present the benefit of using the multiple PR models rather than the single PR model. As aforementioned, the case C is impractical since the case C will cause high $L_p$ distance. Hence, we do not include case C (and D) in this sub-section. From Table 4, the one with the multiple PR models outperforms the one with a single PR model. In average of the accuracy, the multiple PR models produce 1.86% and 1.16% higher than the single PR model for case A and B respectively. This shows the multiple PR models produce a stronger defense than the single PR model. We leave the experiments for the parameters tuning of multiple PR models in the future.

Table 3: Comparison of using non-PR model and PR model (case B) in several datasets with 1,000 epochs under four different attacks.

| Dataset | Model | Accuracy | | | | |
|---------|-------|----------|------|-----|-----|-------|
| | | Normal | FGSM | BIM | MIM | $L_2$-CW |
| MNIST | Non-PR | **0.9893** | 0.1114 | 0.1005 | 0.0844 | 0.1568 |
| | PR (case B), $k$=3 | 0.9870 | **0.8787** | **0.7930** | **0.7504** | **0.6621** |
| Fashion MNIST | Non-PR | **0.8964** | 0.6308 | 0.4221 | 0.2718 | 0.0680 |
| | PR (case B), $k$=3 | 0.8632 | **0.7642** | **0.7271** | **0.6815** | **0.1048** |
| CIFAR-10 | Non-PR | **0.5873** | 0.1807 | 0.1819 | 0.1761 | 0.1612 |
| | PR (case B), $k$=3 | 0.5204 | **0.2265** | **0.2104** | **0.1904** | **0.1728** |
| CIFAR-10 (grayscale) | Non-PR | **0.5369** | 0.1816 | 0.1876 | 0.1839 | 0.1670 |
| | PR (case B), $k$=3 | 0.4569 | **0.2217** | **0.2079** | **0.1888** | **0.1845** |

Table 4: The comparison of using a PR model and multiple PR models in several datasets with 1,000 epochs for the case A and B.

| Dataset | Attack | Accuracy | | | |
|---------|--------|----------|---|---|---|
| | | single PR | | multiple PR | |
| | | Case A | Case B | Case A | Case B |
| MNIST | FGSM, $\epsilon = 0.300$ | 0.9689 | 0.8787 | **0.9752** | **0.9371** |
| | BIM, $\epsilon = 0.300$ | 0.9831 | 0.7930 | **0.9865** | **0.8003** |
| | MIM, $\epsilon = 0.300$ | 0.9680 | 0.7504 | **0.9706** | **0.7693** |
| | $L_2$-CW | 0.9853 | **0.6621** | **0.9875** | 0.5767 |
| Fashion MNIST | FGSM, $\epsilon = 0.031$ | 0.8426 | 0.7642 | **0.8564** | **0.8059** |
| | BIM, $\epsilon = 0.031$ | 0.8420 | 0.7271 | **0.8555** | **0.7703** |
| | MIM, $\epsilon = 0.031$ | 0.8340 | 0.6815 | **0.8501** | **0.7422** |
| | $L_2$-CW | 0.8450 | 0.1048 | **0.8476** | **0.1615** |
| CIFAR-10 | FGSM, $\epsilon = 0.031$ | 0.3343 | **0.2265** | **0.3630** | 0.2181 |
| | BIM, $\epsilon = 0.031$ | 0.3554 | **0.2104** | **0.3859** | 0.2040 |
| | MIM, $\epsilon = 0.031$ | 0.3025 | **0.1904** | **0.3353** | 0.1853 |
| | $L_2$-CW | 0.4667 | **0.1728** | **0.4964** | 0.1717 |
| CIFAR-10 (grayscale) | FGSM, $\epsilon = 0.031$ | 0.3371 | 0.2217 | **0.3655** | **0.2222** |
| | BIM, $\epsilon = 0.031$ | 0.3487 | 0.2079 | **0.3812** | **0.2118** |
| | MIM, $\epsilon = 0.031$ | 0.3216 | 0.1888 | **0.3532** | **0.1948** |
| | $L_2$-CW | 0.4290 | **0.1845** | **0.4526** | 0.1795 |

## 4.4 EXPERIMENT TO ANSWER QUESTION # 4

The aim of conducting these experiments are to distinguish between PR model and random noise injection. Note that the value of the random noise that we used in this sub-section is between -1.0 and 1.0 with the uniform distribution and we clip the value of the pixel between 0.0 and 1.0 (normalized pixel value). We list several outputs with the use of the PR method and random noise injection in Fig. 4.

Besides that, we show the experiment results for the random noises injection with the legitimate data in Table 5. We use a similar settings in Section 4.1 to only apply the random noises injection during the training phase, testing phase and both phases. Comparing the results of the PR method (shown in Table 1) and those of the random noise injection (shown in Table 5), we can see that the PR method outperforms the random noise injection in "Training Phase" and "Both Phases".

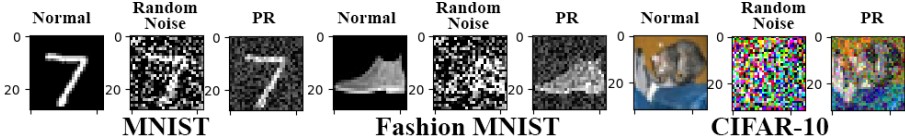

Figure 4: Examples of using PR method and some random noises in MNIST, Fashion MNIST and CIFAR-10 datasets. For every three images from the left, the leftmost one is the original image, the middle one is the image after adding some random noises, and the rightmost one is the PR image.

Although the random noise injection has high performance in fooling the trained neural networks (in the case when we apply the random noise injection during the testing phase only), the $L_2$ distance is larger than our method since the image generated with the random noise injection cannot be easily recognized by the human. (shown in Fig. 4) In summary, the PR method is different from the random noise injection in terms of the performance in maintaining the accuracy when the number of the epoch is increased. Moreover, the PR method can maintain the newly generated image perceptible by the human.

Table 5: Comparison results of applying the some random noises on the image during the different phase.

| Dataset | Epoch | Accuracy | | |
|---|---|---|---|---|
| | | Training Phase | Testing Phase | Both Phases |
| MNIST | 50 | 0.9871 | 0.1650 | 0.9596 |
| | 500 | 0.2789 | 0.1958 | 0.9705 |
| | 1000 | 0.1021 | 0.1695 | 0.9758 |
| Fashion MNIST | 50 | 0.8275 | 0.1435 | 0.8161 |
| | 500 | 0.7308 | 0.1357 | 0.8446 |
| | 1000 | 0.7263 | 0.1297 | 0.8449 |
| CIFAR-10 | 50 | 0.5112 | 0.2205 | 0.4779 |
| | 500 | 0.4296 | 0.1883 | 0.4674 |
| | 1000 | 0.4194 | 0.4739 | 0.4718 |

### 4.5 SUMMARY OF PR METHOD

From the experiment results, we summarize the contribution of using the PR method as follows:

- The PR method relieves the over-fitting issue by generating different possible output of the image. (Q1)

- The PR method increases the robustness of the neural networks.(Q2)

- The defense level for the one with the multiple PR models is stronger than the one with the single PR model. (Q3)

- The PR model has higher performance than the random noise injection. (Q4)

## 5 CONCLUSION

In this study, we propose a novel method, namely Pixel Redrawn method which regenerates the image to all possible forms without compromising human perception. We address four questions in our experiments: **Q1** What is the influence of applying the PR method in different phases with a legitimate test datum?, **Q2** What is the influence of using the PR method in different case studies?, **Q3** What is the difference between multiple PR models and a single PR model applying in the neural networks?, and **Q4** What is the difference between the PR method and the random noise injection? From the experimental results, our method shows that it can prevent the over-fitting problem while maintaining the performance of the neural network, it also improves the robustness of the neural network. Furthermore, our method is more reliable than using some random noises.

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

# A  APPENDIX

## A.1  EXPERIMENTS TO ANSWER QUESTION #2

In this appendix, we provide Table 6, which is the extended version of Table 2 in Section 4.2. Note that more results with 50 and 500 epochs are added. We also provide Table 7 and 8 with $k = 4$ and $k = 10$ respectively. The purpose of the experiments is to analyze the influence of the $k$ value (the number of colors) to the performance of the neural network. From Table 6, 7 and 8, we observe that the bigger the $k$, the higher the accuracy when we test with the legitimate example, but the lower the accuracy when we test with the adversarial examples. In other words, we have to deal with a trade-off between legitimate examples and adversarial examples. Although we believe that multiple PR models could increase the accuracy for adversarial examples while using a large value of $k$, we leave these experiments in the future.

## A.2  EXPERIMENTS TO ANSWER QUESTION #3

In this appendix, we show Table 10 as the extended version of Table 4 in Section 4.3 with the results of the multiple PR models and the single PR model with two epochs (i.e. 500 and 1,000 epochs). In Table 10, it can be seen that the multiple PR models outperforms the single PR model, because the multiple PR models provide more new images to train as was discussed in Section 2.5.

## A.3  EXPERIMENTS TO ANSWER QUESTION #4

In this appendix, we show the comparison results between PR method (case A) and adversarial training with 1,000 epochs for several datasets in Table 11. We apply MIM attack technique during the adversarial training. In Table 11, it can be seen that our method outperforms adversarial training in most cases.

Table 6: Results for several datasets in the case A, B, C, and D with 50, 500, and 1000 epochs and $k$=3. Note that in the case A, we classify as $F(PR(A(F,X)))$. In the case B, we classify as $F(A(F,X))$. In the case C, we classify as $F(A(F,PR,X))$. In the case D, we classify as $F(PR(A(F,PR,X)))$.

| Dataset | Attack | Epoch | Accuracy, $k$=3 | | | | |
|---|---|---|---|---|---|---|---|
| | | | Normal | Case A | Case B | Case C | Case D |
| MNIST | FGSM, $\epsilon = 0.300$ | 50 | 0.9897 | 0.9570 | 0.8527 | 0.4187 | 0.3413 |
| | | 500 | 0.9867 | 0.9698 | 0.9045 | 0.5870 | 0.3919 |
| | | 1000 | 0.9870 | 0.9689 | 0.8787 | 0.6396 | 0.3664 |
| | BIM, $\epsilon = 0.300$ | 50 | 0.9897 | 0.9601 | 0.4485 | 0.0260 | 0.2230 |
| | | 500 | 0.9867 | 0.9786 | 0.8003 | 0.3982 | 0.7223 |
| | | 1000 | 0.9870 | 0.9831 | 0.7930 | 0.5115 | 0.7697 |
| | MIM, $\epsilon = 0.300$ | 50 | 0.9897 | 0.9450 | 0.3651 | 0.0119 | 0.1043 |
| | | 500 | 0.9867 | 0.9636 | 0.7574 | 0.3272 | 0.4788 |
| | | 1000 | 0.9870 | 0.9680 | 0.7504 | 0.3956 | 0.5209 |
| | $L_2$-CW | 50 | 0.9897 | 0.9824 | 0.1617 | 0.0079 | 0.9666 |
| | | 500 | 0.9867 | 0.9842 | 0.6758 | 0.0837 | 0.9724 |
| | | 1000 | 0.9870 | 0.9853 | 0.6621 | 0.1526 | 0.9690 |
| Fashion MNIST | FGSM, $\epsilon = 0.031$ | 50 | 0.8724 | 0.8291 | 0.7366 | 0.6001 | 0.6811 |
| | | 500 | 0.8670 | 0.8380 | 0.7492 | 0.6748 | 0.7376 |
| | | 1000 | 0.8632 | 0.8426 | 0.7642 | 0.6887 | 0.7430 |
| | BIM, $\epsilon = 0.031$ | 50 | 0.8724 | 0.8215 | 0.6944 | 0.4886 | 0.6578 |
| | | 500 | 0.8670 | 0.8360 | 0.7038 | 0.5899 | 0.7229 |
| | | 1000 | 0.8632 | 0.8420 | 0.7271 | 0.6277 | 0.7376 |
| | MIM, $\epsilon = 0.031$ | 50 | 0.8724 | 0.8186 | 0.6662 | 0.4510 | 0.6312 |
| | | 500 | 0.8670 | 0.8302 | 0.6644 | 0.5484 | 0.6998 |
| | | 1000 | 0.8632 | 0.8340 | 0.6815 | 0.5847 | 0.7137 |
| | $L_2$-CW | 50 | 0.8724 | 0.8292 | 0.0770 | 0.0888 | 0.8170 |
| | | 500 | 0.8670 | 0.8444 | 0.0826 | 0.0826 | 0.8196 |
| | | 1000 | 0.8632 | 0.8450 | 0.1048 | 0.0932 | 0.8159 |
| CIFAR-10 | FGSM, $\epsilon = 0.031$ | 50 | 0.5386 | 0.2968 | 0.1931 | 0.1893 | 0.2103 |
| | | 500 | 0.5323 | 0.3256 | 0.2053 | 0.2001 | 0.2186 |
| | | 1000 | 0.5204 | 0.3343 | 0.2265 | 0.2155 | 0.2236 |
| | BIM, $\epsilon = 0.031$ | 50 | 0.5386 | 0.3127 | 0.1876 | 0.1865 | 0.2038 |
| | | 500 | 0.5323 | 0.3385 | 0.1932 | 0.1957 | 0.2114 |
| | | 1000 | 0.5204 | 0.3554 | 0.2104 | 0.2054 | 0.2216 |
| | MIM, $\epsilon = 0.031$ | 50 | 0.5386 | 0.2772 | 0.1783 | 0.1815 | 0.1920 |
| | | 500 | 0.5323 | 0.2898 | 0.1775 | 0.1839 | 0.1916 |
| | | 1000 | 0.5204 | 0.3025 | 0.1904 | 0.1913 | 0.1938 |
| | $L_2$-CW | 50 | 0.5386 | 0.4728 | 0.1646 | 0.1687 | 0.4256 |
| | | 500 | 0.5323 | 0.4708 | 0.1626 | 0.1765 | 0.4234 |
| | | 1000 | 0.5204 | 0.4667 | 0.1728 | 0.1654 | 0.4156 |
| CIFAR-10 (grayscale) | FGSM, $\epsilon = 0.031$ | 50 | 0.4845 | 0.3430 | 0.2216 | 0.2002 | 0.2285 |
| | | 500 | 0.4609 | 0.3357 | 0.2098 | 0.1960 | 0.2472 |
| | | 1000 | 0.4569 | 0.3371 | 0.2217 | 0.2016 | 0.2502 |
| | BIM, $\epsilon = 0.031$ | 50 | 0.4845 | 0.3518 | 0.2112 | 0.1910 | 0.2387 |
| | | 500 | 0.4609 | 0.3494 | 0.2005 | 0.1924 | 0.2531 |
| | | 1000 | 0.4569 | 0.3451 | 0.2079 | 0.1895 | 0.2633 |
| | MIM, $\epsilon = 0.031$ | 50 | 0.4845 | 0.3270 | 0.1971 | 0.1851 | 0.2165 |
| | | 500 | 0.4609 | 0.3181 | 0.1834 | 0.1819 | 0.2289 |
| | | 1000 | 0.4569 | 0.3158 | 0.1888 | 0.1808 | 0.2291 |
| | $L_2$-CW | 50 | 0.4845 | 0.4424 | 0.1817 | 0.1788 | 0.3865 |
| | | 500 | 0.4609 | 0.4363 | 0.1727 | 0.1719 | 0.3945 |
| | | 1000 | 0.4569 | 0.4290 | 0.1845 | 0.1727 | 0.3868 |

Table 7: Results for several datasets in the case A, B, C, and D with 50, 500, and 1000 epochs and $k$=4. Note that in the case A, we classify as $F(PR(A(F, X)))$. In the case B, we classify as $F(A(F, X))$. In the case C, we classify as $F(A(F, PR, X))$. In the case D, we classify as $F(PR(A(F, PR, X)))$.

| Dataset | Attack | Epoch | Accuracy, $k$=4 | | | | |
| --- | --- | --- | --- | --- | --- | --- | --- |
| | | | Normal | Case A | Case B | Case C | Case D |
| MNIST | FGSM, $\epsilon = 0.300$ | 50 | 0.9932 | 0.7421 | 0.7908 | 0.5384 | 0.5502 |
| | | 500 | 0.9891 | 0.8596 | 0.8756 | 0.8131 | 0.8238 |
| | | 1000 | 0.9908 | 0.8114 | 0.8409 | 0.7198 | 0.6961 |
| | BIM, $\epsilon = 0.300$ | 50 | 0.9932 | 0.5126 | 0.3372 | 0.0810 | 0.1680 |
| | | 500 | 0.9891 | 0.9144 | 0.7817 | 0.6947 | 0.8227 |
| | | 1000 | 0.9908 | 0.9283 | 0.6446 | 0.4903 | 0.7386 |
| | MIM, $\epsilon = 0.300$ | 50 | 0.9932 | 0.3238 | 0.2487 | 0.0517 | 0.0978 |
| | | 500 | 0.9891 | 0.7932 | 0.7231 | 0.6441 | 0.7019 |
| | | 1000 | 0.9908 | 0.6979 | 0.5380 | 0.3920 | 0.4730 |
| | $L_2$-CW | 50 | 0.9932 | 0.9885 | 0.2584 | 0.0471 | 0.9914 |
| | | 500 | 0.9891 | 0.9881 | 0.8142 | 0.7988 | 0.9892 |
| | | 1000 | 0.9908 | 0.9902 | 0.8437 | 0.8034 | 0.9906 |
| Fashion MNIST | FGSM, $\epsilon = 0.031$ | 50 | 0.8896 | 0.8132 | 0.7363 | 0.6531 | 0.7836 |
| | | 500 | 0.8841 | 0.8450 | 0.8071 | 0.7671 | 0.8220 |
| | | 1000 | 0.8830 | 0.8507 | 0.8212 | 0.7761 | 0.8242 |
| | BIM, $\epsilon = 0.031$ | 50 | 0.8896 | 0.7864 | 0.6371 | 0.5140 | 0.7626 |
| | | 500 | 0.8841 | 0.8330 | 0.7419 | 0.6536 | 0.8170 |
| | | 1000 | 0.8830 | 0.8463 | 0.7913 | 0.7207 | 0.8248 |
| | MIM, $\epsilon = 0.031$ | 50 | 0.8896 | 0.7792 | 0.6253 | 0.4984 | 0.7477 |
| | | 500 | 0.8841 | 0.8280 | 0.7232 | 0.6277 | 0.8070 |
| | | 1000 | 0.8830 | 0.8419 | 0.7622 | 0.6811 | 0.8020 |
| | $L_2$-CW | 50 | 0.8896 | 0.8424 | 0.0739 | 0.0746 | 0.8759 |
| | | 500 | 0.8841 | 0.8430 | 0.1345 | 0.1217 | 0.8677 |
| | | 1000 | 0.8830 | 0.8501 | 0.1590 | 0.1650 | 0.8668 |
| CIFAR-10 | FGSM, $\epsilon = 0.031$ | 50 | 0.5969 | 0.2398 | 0.1889 | 0.1902 | 0.2038 |
| | | 500 | 0.5838 | 0.2667 | 0.2020 | 0.1983 | 0.2199 |
| | | 1000 | 0.5764 | 0.2809 | 0.2254 | 0.2132 | 0.2391 |
| | BIM, $\epsilon = 0.031$ | 50 | 0.5969 | 0.2397 | 0.1820 | 0.1881 | 0.1993 |
| | | 500 | 0.5838 | 0.2682 | 0.1916 | 0.1904 | 0.2078 |
| | | 1000 | 0.5764 | 0.2887 | 0.2072 | 0.1985 | 0.2253 |
| | MIM, $\epsilon = 0.031$ | 50 | 0.5969 | 0.2126 | 0.1760 | 0.1828 | 0.1890 |
| | | 500 | 0.5838 | 0.2203 | 0.1755 | 0.1789 | 0.1857 |
| | | 1000 | 0.5764 | 0.2347 | 0.1851 | 0.1812 | 0.1961 |
| | $L_2$-CW | 50 | 0.5969 | 0.4758 | 0.1626 | 0.1681 | 0.4661 |
| | | 500 | 0.5838 | 0.4716 | 0.1645 | 0.1696 | 0.4573 |
| | | 1000 | 0.5764 | 0.4668 | 0.1742 | 0.1705 | 0.4595 |
| CIFAR-10 (grayscale) | FGSM, $\epsilon = 0.031$ | 50 | 0.5362 | 0.2805 | 0.1964 | 0.1904 | 0.2344 |
| | | 500 | 0.5316 | 0.3061 | 0.1999 | 0.1999 | 0.2501 |
| | | 1000 | 0.5210 | 0.3076 | 0.2079 | 0.2027 | 0.2588 |
| | BIM, $\epsilon = 0.031$ | 50 | 0.5362 | 0.2860 | 0.1964 | 0.1839 | 0.2435 |
| | | 500 | 0.5316 | 0.3135 | 0.1999 | 0.1892 | 0.2591 |
| | | 1000 | 0.5210 | 0.3191 | 0.2079 | 0.1928 | 0.2663 |
| | MIM, $\epsilon = 0.031$ | 50 | 0.5362 | 0.2571 | 0.1864 | 0.1802 | 0.2218 |
| | | 500 | 0.5316 | 0.2679 | 0.1830 | 0.1773 | 0.2221 |
| | | 1000 | 0.5210 | 0.2719 | 0.1903 | 0.1813 | 0.2238 |
| | $L_2$-CW | 50 | 0.5362 | 0.4710 | 0.1762 | 0.1672 | 0.4543 |
| | | 500 | 0.5316 | 0.4642 | 0.1668 | 0.1693 | 0.4557 |
| | | 1000 | 0.5210 | 0.4595 | 0.1763 | 0.1680 | 0.4494 |

Table 8: Results for several datasets in the case A, B, C, and D with 50, 500, and 1000 epochs and $k$=10. Note that in the case A, we classify as $F(PR(A(F, X)))$. In the case B, we classify as $F(A(F, X))$. In the case C, we classify as $F(A(F, PR, X))$. In the case D, we classify as $F(PR(A(F, PR, X)))$.

| Dataset | Attack | Epoch | Accuracy, $k$=10 | | | | |
| | | | Normal | Case A | Case B | Case C | Case D |
| MNIST | FGSM, $\epsilon = 0.300$ | 50 | 0.9933 | 0.8324 | 0.6713 | 0.4700 | 0.6466 |
| | | 500 | 0.9897 | 0.8754 | 0.7677 | 0.7641 | 0.8184 |
| | | 1000 | 0.9890 | 0.8196 | 0.7500 | 0.6874 | 0.6644 |
| | BIM, $\epsilon = 0.300$ | 50 | 0.9933 | 0.4206 | 0.1431 | 0.0681 | 0.1809 |
| | | 500 | 0.9897 | 0.8125 | 0.5887 | 0.5203 | 0.7374 |
| | | 1000 | 0.9890 | 0.8583 | 0.5935 | 0.5968 | 0.7864 |
| | MIM, $\epsilon = 0.300$ | 50 | 0.9933 | 0.2076 | 0.0758 | 0.0333 | 0.0693 |
| | | 500 | 0.9897 | 0.5491 | 0.4203 | 0.3167 | 0.3952 |
| | | 1000 | 0.9890 | 0.5912 | 0.4350 | 0.3323 | 0.3881 |
| | $L_2$-CW | 50 | 0.9933 | 0.9906 | 0.0877 | 0.0231 | 0.9928 |
| | | 500 | 0.9897 | 0.9892 | 0.8272 | 0.5395 | 0.9896 |
| | | 1000 | 0.9890 | 0.9877 | 0.4854 | 0.4725 | 0.9880 |
| Fashion MNIST | FGSM, $\epsilon = 0.031$ | 50 | 0.8957 | 0.7773 | 0.6876 | 0.6381 | 0.7889 |
| | | 500 | 0.8864 | 0.8156 | 0.7825 | 0.7558 | 0.8172 |
| | | 1000 | 0.8802 | 0.8257 | 0.8029 | 0.7700 | 0.8188 |
| | BIM, $\epsilon = 0.031$ | 50 | 0.8957 | 0.7282 | 0.5497 | 0.4788 | 0.7643 |
| | | 500 | 0.8864 | 0.7935 | 0.7055 | 0.6570 | 0.8142 |
| | | 1000 | 0.8802 | 0.8265 | 0.7712 | 0.7141 | 0.8288 |
| | MIM, $\epsilon = 0.031$ | 50 | 0.8957 | 0.7154 | 0.5449 | 0.4728 | 0.7531 |
| | | 500 | 0.8864 | 0.7753 | 0.6710 | 0.6124 | 0.7985 |
| | | 1000 | 0.8802 | 0.8147 | 0.7241 | 0.6501 | 0.8177 |
| | $L_2$-CW | 50 | 0.8957 | 0.8369 | 0.0681 | 0.0714 | 0.8854 |
| | | 500 | 0.8864 | 0.8185 | 0.1400 | 0.1374 | 0.8682 |
| | | 1000 | 0.8802 | 0.8282 | 0.1532 | 0.1475 | 0.8633 |
| CIFAR-10 | FGSM, $\epsilon = 0.031$ | 50 | 0.6087 | 0.1928 | 0.1855 | 0.1808 | 0.1883 |
| | | 500 | 0.5989 | 0.1945 | 0.1836 | 0.1840 | 0.1936 |
| | | 1000 | 0.5918 | 0.2006 | 0.1906 | 0.1903 | 0.2022 |
| | BIM, $\epsilon = 0.031$ | 50 | 0.6087 | 0.1892 | 0.1822 | 0.1795 | 0.1885 |
| | | 500 | 0.5989 | 0.1901 | 0.1801 | 0.1819 | 0.1951 |
| | | 1000 | 0.5918 | 0.1971 | 0.1868 | 0.1828 | 0.2013 |
| | MIM, $\epsilon = 0.031$ | 50 | 0.6087 | 0.1840 | 0.1797 | 0.1766 | 0.1780 |
| | | 500 | 0.5989 | 0.1766 | 0.1742 | 0.1754 | 0.1815 |
| | | 1000 | 0.5918 | 0.1819 | 0.1781 | 0.1763 | 0.1811 |
| | $L_2$-CW | 50 | 0.6087 | 0.3658 | 0.1591 | 0.1608 | 0.3792 |
| | | 500 | 0.5989 | 0.3967 | 0.1672 | 0.1678 | 0.4038 |
| | | 1000 | 0.5918 | 0.3884 | 0.1665 | 0.1640 | 0.4063 |
| CIFAR-10 (grayscale) | FGSM, $\epsilon = 0.031$ | 50 | 0.5626 | 0.1973 | 0.1895 | 0.1886 | 0.1941 |
| | | 500 | 0.5566 | 0.2012 | 0.1872 | 0.1834 | 0.1894 |
| | | 1000 | 0.5545 | 0.2126 | 0.1949 | 0.1923 | 0.2027 |
| | BIM, $\epsilon = 0.031$ | 50 | 0.5626 | 0.1980 | 0.1857 | 0.1865 | 0.1932 |
| | | 500 | 0.5566 | 0.1945 | 0.1799 | 0.1800 | 0.1929 |
| | | 1000 | 0.5545 | 0.2083 | 0.1869 | 0.1850 | 0.1982 |
| | MIM, $\epsilon = 0.031$ | 50 | 0.5626 | 0.1890 | 0.1813 | 0.1816 | 0.1857 |
| | | 500 | 0.5566 | 0.1761 | 0.1704 | 0.1733 | 0.1778 |
| | | 1000 | 0.5545 | 0.1841 | 0.1757 | 0.1757 | 0.1801 |
| | $L_2$-CW | 50 | 0.5626 | 0.4075 | 0.1664 | 0.1689 | 0.4189 |
| | | 500 | 0.5566 | 0.4206 | 0.1594 | 0.1591 | 0.4373 |
| | | 1000 | 0.5545 | 0.4247 | 0.1656 | 0.1629 | 0.4336 |

Table 9: Comparison of using non-PR model and PR model (case B) in several datasets with 50, 500, and 1,000 epochs under four different attacks.

| Dataset | Attack | Epoch | Accuracy | |
|---|---|---|---|---|
| | | | non-PR | PR (case B), $k=3$ |
| MNIST | Normal | 50 | **0.9921** | 0.9897 |
| | | 500 | **0.9885** | 0.9867 |
| | | 1000 | **0.9893** | 0.9870 |
| | FGSM, $\epsilon = 0.300$ | 50 | 0.2004 | **0.8527** |
| | | 500 | 0.2187 | **0.9045** |
| | | 1000 | 0.1114 | **0.8787** |
| | BIM, $\epsilon = 0.300$ | 50 | 0.0369 | **0.4485** |
| | | 500 | 0.0576 | **0.8003** |
| | | 1000 | 0.1005 | **0.7930** |
| | MIM, $\epsilon = 0.300$ | 50 | 0.0230 | **0.3651** |
| | | 500 | 0.0549 | **0.7574** |
| | | 1000 | 0.0844 | **0.7504** |
| | $L_2$-CW | 50 | 0.0156 | **0.1617** |
| | | 500 | 0.1018 | **0.6758** |
| | | 1000 | 0.1568 | **0.6621** |
| Fashion MNIST | Normal | 50 | **0.9106** | 0.8724 |
| | | 500 | **0.9058** | 0.8670 |
| | | 1000 | **0.8964** | 0.8632 |
| | FGSM, $\epsilon = 0.031$ | 50 | 0.6083 | **0.7366** |
| | | 500 | 0.6624 | **0.7492** |
| | | 1000 | 0.6308 | **0.7642** |
| | BIM, $\epsilon = 0.031$ | 50 | 0.4265 | **0.6944** |
| | | 500 | 0.4064 | **0.7038** |
| | | 1000 | 0.4221 | **0.7271** |
| | MIM, $\epsilon = 0.031$ | 50 | 0.3566 | **0.6662** |
| | | 500 | 0.2401 | **0.6644** |
| | | 1000 | 0.2718 | **0.6815** |
| | $L_2$-CW | 50 | 0.0647 | **0.0700** |
| | | 500 | 0.0652 | **0.0826** |
| | | 1000 | 0.0680 | **0.1048** |
| CIFAR-10 | Normal | 50 | **0.6179** | 0.5386 |
| | | 500 | **0.5989** | 0.5323 |
| | | 1000 | **0.5873** | 0.5204 |
| | FGSM, $\epsilon = 0.031$ | 50 | 0.1801 | **0.1931** |
| | | 500 | 0.1753 | **0.2053** |
| | | 1000 | 0.1807 | **0.2265** |
| | BIM, $\epsilon = 0.031$ | 50 | 0.1767 | **0.1876** |
| | | 500 | 0.1751 | **0.1932** |
| | | 1000 | 0.1819 | **0.2104** |
| | MIM, $\epsilon = 0.031$ | 50 | 0.1741 | **0.1783** |
| | | 500 | 0.1723 | **0.1775** |
| | | 1000 | 0.1761 | **0.1904** |
| | $L_2$-CW | 50 | 0.1603 | **0.1646** |
| | | 500 | 0.1576 | **0.1626** |
| | | 1000 | 0.1612 | **0.1728** |
| CIFAR-10 (grayscale) | Normal | 50 | **0.5697** | 0.4845 |
| | | 500 | **0.5466** | 0.4609 |
| | | 1000 | **0.5369** | 0.4569 |
| | FGSM, $\epsilon = 0.031$ | 50 | 0.1781 | **0.2216** |
| | | 500 | 0.1732 | **0.2098** |
| | | 1000 | 0.1816 | **0.2217** |
| | BIM, $\epsilon = 0.031$ | 50 | 0.1786 | **0.2112** |
| | | 500 | 0.1768 | **0.2005** |
| | | 1000 | 0.1876 | **0.2079** |
| | MIM, $\epsilon = 0.031$ | 50 | 0.1745 | **0.1971** |
| | | 500 | 0.1744 | **0.1834** |
| | | 1000 | 0.1839 | **0.1888** |
| | $L_2$-CW | 50 | 0.1597 | **0.1817** |
| | | 500 | 0.1621 | **0.1727** |
| | | 1000 | 0.1670 | **0.1845** |

Table 10: The comparison of using a PR model and multiple PR models in several datasets with 500 and 1,000 epochs respectively, for case A and case B.

| Dataset | Attack | Epoch | Accuracy | | | |
|---|---|---|---|---|---|---|
| | | | single PR | | multiple PR | |
| | | | Case A | Case B | Case A | Case B |
| MNIST, $\epsilon = 0.300$ | FGSM | 500 | 0.9698 | 0.9045 | **0.9724** | **0.9350** |
| | | 1000 | 0.9698 | 0.8787 | **0.9752** | **0.9371** |
| | BIM | 500 | 0.9786 | 0.8003 | **0.9816** | **0.8224** |
| | | 1000 | 0.9831 | 0.7930 | **0.9865** | **0.8003** |
| | MIM | 500 | 0.9636 | 0.7574 | **0.9645** | **0.7806** |
| | | 1000 | 0.9680 | 0.7504 | **0.9706** | **0.7693** |
| | $L_2$-CW | 500 | 0.9842 | **0.6758** | **0.9872** | 0.3273 |
| | | 1000 | 0.9853 | **0.6621** | **0.9875** | 0.5767 |
| Fashion MNIST, $\epsilon = 0.031$ | FGSM | 500 | 0.8380 | 0.7492 | **0.8582** | **0.7903** |
| | | 1000 | 0.8426 | 0.7642 | **0.8564** | **0.8059** |
| | BIM | 500 | 0.8360 | 0.7038 | **0.8527** | **0.7575** |
| | | 1000 | 0.8420 | 0.7271 | **0.8555** | **0.7703** |
| | MIM | 500 | 0.8302 | 0.6644 | **0.8503** | **0.7310** |
| | | 1000 | 0.8340 | 0.6815 | **0.8501** | **0.7422** |
| | $L_2$-CW | 500 | 0.8444 | 0.0826 | **0.8487** | **0.1432** |
| | | 1000 | 0.8450 | 0.1048 | **0.8476** | **0.1615** |
| CIFAR-10, $\epsilon = 0.031$ | FGSM | 500 | 0.3256 | 0.2053 | **0.3470** | **0.2104** |
| | | 1000 | 0.3343 | **0.2265** | **0.3630** | 0.2181 |
| | BIM | 500 | 0.3385 | 0.1932 | **0.3689** | **0.2013** |
| | | 1000 | 0.3554 | **0.2104** | **0.3859** | 0.2040 |
| | MIM | 500 | 0.2898 | 0.1775 | **0.3189** | **0.1847** |
| | | 1000 | 0.3025 | **0.1904** | **0.3353** | 0.1853 |
| | $L_2$-CW | 500 | 0.4708 | 0.1626 | **0.4947** | **0.1709** |
| | | 1000 | 0.4667 | **0.1728** | **0.4964** | 0.1717 |
| CIFAR-10 (grayscale), $\epsilon = 0.031$ | FGSM | 500 | 0.3357 | 0.2098 | **0.3665** | **0.2174** |
| | | 1000 | 0.3371 | 0.2217 | **0.3655** | **0.2222** |
| | BIM | 500 | 0.3534 | 0.2005 | **0.3750** | **0.2083** |
| | | 1000 | 0.3487 | 0.2079 | **0.3812** | **0.2118** |
| | MIM | 500 | 0.3234 | 0.1834 | **0.3526** | **0.1917** |
| | | 1000 | 0.3216 | 0.1888 | **0.3532** | **0.1948** |
| | $L_2$-CW | 500 | 0.4363 | 0.1727 | **0.4483** | **0.1829** |
| | | 1000 | 0.4290 | **0.1845** | **0.4526** | 0.1795 |

Table 11: Comparison between PR method (case A) and adversarial training in several datasets with 1,000 epochs.

| Dataset | Attack | Accuracy | |
|---|---|---|---|
| | | PR | Adversarial Training |
| MNIST, $\epsilon = 0.300$ | FGSM | 0.9691 | **0.9712** |
| | BIM | **0.9825** | 0.9264 |
| | MIM | **0.9672** | 0.9287 |
| Fashion MNIST, $\epsilon = 0.031$ | FGSM | **0.8503** | 0.8408 |
| | BIM | **0.8489** | 0.8190 |
| | MIM | **0.8446** | 0.8059 |
| CIFAR-10, $\epsilon = 0.031$ | FGSM | 0.3343 | **0.3830** |
| | BIM | **0.3554** | 0.3495 |
| | MIM | 0.3025 | **0.3072** |

