# OpenReview forum: "Pixel Redrawn For A Robust Adversarial Defense"
_ICLR.cc/2019/Conference_

### Official Review · AnonReviewer3 · 2018-11-02
**Results are not convincing**

**Rating:** 3
**Confidence:** 4

**Review:**

This paper presents a new adversarial defense method.  I found the paper difficult to understand, but as far as I understand, the method involves randomly perturbing the pixels of images in a dataset, and retraining the classifier to correctly classify these perturbed images as well.  The perturbations are done independently per pixel by quantizing the pixel value, and then using a perceptron model to generate the full pixel value from just this quantized value.  The perceptron is trained on randomly generated data mapping quantized pixel values to full pixel values (this random generation does not use the dataset statistics at all).  They use both partially converged perceptron models as well as fully converged perceptron models.

Pros:
	1. The defense technique does not require knowledge of the attack method

Cons:
	1. The paper is incredibly difficult to understand due to the writing.
	2. The performed experiments are insufficient to determine  whether or not their technique works because:
		a. They don't compare against any other defense techniques.  In addition to at least adversarial training, I would like to also see comparisons to feature squeezing which I would expect to have a very similar effect.
		b. They do not show the results of an attack by an adversary that is aware of their technique.  (i.e. F(PR(A(F,PR,x))))  Many alternative defense techniques will work much better if we assume the adversary does not know about the technique.
		c. Their comparison against random noise is not an apples-to-apples comparison.  Instead of perturbing uniformly within a range, they perturb according to a normal distribution.  The random noise perturbations also have a much larger L2 distance than the perturbations from their technique.  To believe that their method is actually better than training with random noise I'd like to see an apples-to-apples comparison where these values are hyperparameter tuned as they presumably did for their method.
		d. They only show results for one value of epsilon and one value of "# of colors" for their technique.  Presumably if there is a mismatch between these values then the results will be much worse (i.e. if they choose a large "# of colors"/small range per color and the attacker chooses a large epsilon).
	3. Their use of machine learning models is quite ad-hoc.  In particular they use a perceptron trained on algorithmically generated data.  And their justification for using a perceptron, instead of just using the known underlying generation algorithm is that they also use a partially converged version of the perceptron as part of their model.
	4. The authors partly deanonymize the paper through a github link
	5. The main paper is 10 pages long, and the quality of the paper does not justify the additional length.

---

> ### Author Response · Authors · 2018-11-26
> **Reply to "Results are not convincing"**
>
> Thank you for your review and comments.
>
> 1. We have proof-read and added preliminaries (Section 2.1) as well as the clear explanation of our method in the figure format (Figure 1) in this latest version. We believe that the readers could understand our method easily.
>
> 2. a) Following your comment, we have added the comparison with the adversarial training in Appendix section (Section A.3). The comparison results are shown in Table 11. In Table 11, our PR method outperforms the adversarial training which has applied with momentum iterative method (MIM) to generate the adversarial images as the training dataset. We would like to include feature squeezing next time.
>
> 2. b) Following your comment, we have included F(PR(A(F, PR, X))) as Case D in Section 3.4. The experimental results are shown in Table 2. We have also included the full version of the results with different number of epochs in Table 6, Section A.1.
>
> 2. c) We are sorry for the mistake that we have made in the previous version of our manuscript. The distribution that we have used for the random noise injection was a uniform distribution. It is not from a normal distribution.
>
> 2. d) Following your comments, we have operated some experiments with different “# of colors” which we have defined as k in the latest manuscript. In the previous manuscript, we used k=3. In the latest manuscript, we include k=4 and k=10 which the results are shown in Table 7 and 8 respectively, in Section A.1.
>
> 3. There are two reasons we use the perceptron in the paper.
> The perceptron can be used in parallel way during the training or testing phase.
> We can replace the perceptron with a similar convergence rate but with different weights (and biases) if the perceptron has been revealed by the attacker.
> In other words, the perceptron can be used as a “key” to protect the neural networks from the attacker by replacing a new perceptron. A paper with similar idea can be found in the following link. https://openreview.net/forum?id=HkElFj0qYQ&noteId=HJedRiS5hX
>
> 4. Thanks for reminding us. We have changed the github name to a different name.

---

### Official Review · AnonReviewer2 · 2018-11-02
**simple but effective method in triaining but its utility as a defender not quite clear to me why it works**

**Rating:** 6
**Confidence:** 3

**Review:**

The authors propose a defense technique to make the NN model more robust to adversarial events by redrawing the images and use them for training the model so that the model can prevent future attacks. The idea itself is simple but seems to be effective as shown in Tables 2 and 3.

What I’m missing here is a simple experiment to see the difference in accuracy performance between 1) when using PR for training the model (which is Case B in Table 2) against attacks versus 2) when not using PR for training the model against other attacks (similar to Table 2, Testing phase but using other attack models not PR as an attack model).

It is not quite clear to me why using PR as a defense mechanism helps the NN model.  I see its utility when training the NN model but using it as a defense mechanism is not quite clear why it works.

Minor:
It is not quite clear how the author chose the hyperparameters, maybe by changing those hyperparameter the attacker could have much clever ways to attack the NN model.

---

> ### Author Response · Authors · 2018-11-26
> **Reply to "simple but effective method in training but its utility as a defender not quite clear to me why it works"**
>
> Thank you for your review and comments.
>
> Following your comment, we have added the experiments in Section 4.2. The experiment results are shown in Table 3. From the results, the neural network that has trained with PR image increases the robustness of the neural network although the accuracy of legitimate data has slightly decreased compare with the normal neural network.
>
> During the training phase, PR method generates PR images (distorted images) as the training dataset to increase the robustness of the neural network.
> During the testing phase, however, PR method distorts the adversarial image by eliminating most adversarial pixels of the adversarial image to defend the adversarial attack.

---

### Official Review · AnonReviewer1 · 2018-11-02
**A simple noisy perturbation scheme for improving robustness of CNN**

**Rating:** 4
**Confidence:** 3

**Review:**

This paper proposed the pixel redrawing approach to generate distorted training images to improve the performance of the deep networks, which hopefully can be used to prevent future attacks. The key idea is to randomly perturb the pixel values according pre-defined range and probabilities.


The proposed method is quite simple and is similar to denoising autoencoders in flavor.  My concern is that a pre-defined noisy perturbation may not be general enough to tackle various types of attacks. Ideally the perturbation should take into account properties of the input images, both pixel-wise and structure-wise. Unfortunately the proposed method ignores such information. The performance improvement seems quite limited judging from the results.

---

> ### Author Response · Authors · 2018-11-26
> **Reply to "A simple noisy perturbation scheme for improving robustness of CNN"**
>
> Thank you for your review and comments.
>
> Based on our knowledge, most state-of-the-art attack techniques usually are pixel-wise perturbation. Hence, we focus more on this issue and we came out with a distorted image generated by PR method.
>
> From Table 2, the PR method can eliminate most adversarial pixels from the strongest attack technique, L2-CW attack in Case A and Case D (newly added in Section 3.4) due to the mapping method (newly added in Section 2.2).
> From Table 3 (newly added in Section 4. 2), the neural network that has trained with PR images increases the robustness of the neural network.

---

### Public Comment · (anonymous) · 2018-10-25
**Do not understand Case C**

Could you explain again what you mean by Case C? Here is what I got from it, maybe you could tell me if this is correct:

- Case A: The attacker receives an input x, transforms it to an adversarial example x'=A(F,x), and then Pixel Redraw performs PR(x') and the network classifies this image: F(PR(x')). The attacker is assumed to know nothing about the function PR, but knows the function F.

- Case B: The attacker receives an input x, transforms it to an adversarial example A(F,x), and then the network classifies F(A(F,x)). The attacker knows F. In this case PR is not used at all.

- Case C: Is this identical to Case A except now we assume the attacker knows PR? That is, we classify F(PR(A(F,PR,x)))? Or is something else intended here?

---

> ### Author Response · Authors · 2018-10-26
> **Reply to "Do not understand Case C"**
>
> Thank you for your comments.
>
> For Case A and B, they are correct. For Case C, we classify F(A(F, PR(x))).
>
> In detail, we perform as follows:
> 1. x* = PR(x)
> 2. x' = A(F, x*)
> 3. y = F(x')  // without PR method
>
> ** Thanks to the comment, we would like to test the classification, F(PR(A(F,PR,x))) as another case.

---

> > ### Public Comment · (anonymous) · 2018-10-26
> > **Why is this interesting?**
> >
> > Why is Case C interesting then? Why would the adversary ever go *after* the PR function has already operated on the inputs? (The paper says that there is a clearly larger distortion in Case C. Given how you actually run this case, it seems obvious the distortion would be much larger -- you run the attack after adding a very large perturbation with PR.)

---

> > > ### Author Response · Authors · 2018-10-27
> > > **Reply to "Why is this interesting?"**
> > >
> > > Thank you for the comments.
> > > Although we classify F(A(F, PR(x))) for Case C, the results (the accuracy and the distance metric -- adversarial image) are similar to F(A(F, PR, x)).
> > > We show Case C is because we want to evaluate the performance of our defense technique if our defense technique has exposed. Unfortunately, due to our defense technique, the adversarial image generated will have high distortion value. Therefore, we mentioned that Case C is impractical in our paper.
> > > We will add the result of the classifications (i.e. F(A(F, PR, x)) and F(PR(A(F, PR, x)))) in the next version.

---

> > > > ### Public Comment · (anonymous) · 2018-10-27
> > > > **Still don't understand**
> > > >
> > > > I still don't see how this makes sense. Why should the adversary modify the image *after* it has already been modified by PR? In what threat model does this make sense?
> > > >
> > > > (As the defender, when I have a pre-processor, I take some image x and get to do whatever I want to it. With PR, I will just run the function F(PR(x)). It doesn't make sense to assume the adversary some how gets to act in between those two functions.)

---

> > > > > ### Author Response · Authors · 2018-10-28
> > > > > **Reply to "Still don't understand"**
> > > > >
> > > > > Thank you for the comments.
> > > > > Assume that the attackers do not interest with the pre-processing
> > > > > method, either the attackers have knowledge about the pre-processing
> > > > > method (white-box attack) or not (grey-box attack), the results are
> > > > > shown in Case A.
> > > > > If the pre-processing method assigns an pixel with a random number,
> > > > > the method can be defined as one of the following:
> > > > > a) X_new = X_old + n
> > > > > b) X_new = X_old * n
> > > > > c) X_new = X_old * k + n
> > > > > ** X_new = f(X_old)
> > > > >
> > > > > If the neural network is as follows:
> > > > > y = (.)W+b
> > > > > where (.) is the pre-processing method
> > > > >
> > > > > If the attack technique is as follows:
> > > > > X_adv = X_old + eps*sign(dL/dX_old)
> > > > >
> > > > > If the scenario is given as above, the X_adv will produce the same
> > > > > result for the (a) and (b) cases of the pre-processing method. The
> > > > > X_adv may have slightly different in the (c) case of the
> > > > > pre-processing method.
> > > > > In summary, the attackers cannot do much with the pre-processing
> > > > > method (e.g. randomness function) even they know the pre-processing
> > > > > method.
> > > > >
> > > > > On the other hand, if the attackers do want to deal with the
> > > > > pre-processing method, Case C maybe needed since the attackers do not
> > > > > want the adversarial pixels to be removed from the pre-processing
> > > > > method. As far as our knowledge, no much attack techniques will deal
> > > > > with the pre-processing method. This may be an open problem for
> > > > > researchers to study how the attackers can deal with the
> > > > > pre-processing method while maintaining the quality of the adversarial
> > > > > image as good as the original image and the generated adversarial
> > > > > image can fool the trained neural networks.
> > > > >
> > > > > For case C,
> > > > > the neural network and the attack technique are as follows:
> > > > > y = (X_old -> X_new)W+b
> > > > > X_adv = X_new + eps*sign(dL/dX_new)
> > > > >
> > > > > In the python script (tensorflow) that we have implemented, when we
> > > > > apply the classification, F(A(F, PR, X)), where the attackers know all
> > > > > the parameters of the neural network and PR method, the produced
> > > > > results are similar as F(A(F, PR(X))).
> > > > > We conjecture that the derivation of the "X_old -> X_new" is hard to
> > > > > be computed by the script, so, it has stopped at the "X_new" and
> > > > > updated the "X_new" as the adversarial image.
> > > > >
> > > > > We show Case C is also to evaluate the performance of the attack
> > > > > technique and defense technique. From Case C, it showed that attack
> > > > > technique can intrude the defense technique easily but it caused the
> > > > > high distortion of the image

---

> > > > > > ### Public Comment · (anonymous) · 2018-10-29
> > > > > > **Question remains**
> > > > > >
> > > > > > Thank you for your reply. However I don't think this answers my original question.
> > > > > >
> > > > > > You say that one of the cases you study is F(A(F, PR(x))). I do not understand why that case is ever interesting.
> > > > > >
> > > > > > Can you please give an example where (1) the PR method transforms an input x, (2) the attacker modifies the output of the PR() method, and (3) the result of that is then classified by F?
> > > > > >
> > > > > > I can not think of a reason why this case is useful. It doesn't make sense why the attacker should be allowed to take their turn in between running PR() and F()

---

> > > > > > > ### Author Response · Authors · 2018-10-29
> > > > > > > **Reply to "Question remains"**
> > > > > > >
> > > > > > > Thank you for your comments.
> > > > > > >
> > > > > > > The example for each query is shown as follows:
> > > > > > > Assume that X is an image, x_1, x_2, ..., x_n is the pixels of the image.
> > > > > > > (1) For a pixel, x_1 = 20, PR method will transform x_1 to any random value, for example, 28.
> > > > > > > (2) The attacker will use x_1 = 28 to generate a new value, maybe x_1 has changed to 26.
> > > > > > > (3) The simple linear equation for predicting the adversarial image is
> > > > > > >     y = x_1*w_1 + x_2*w_2 + ... + x_n * w_n
> > > > > > >       = 26*w_1 + x_2*w_2 + ... + x_n * w_n
> > > > > > >
> > > > > > > Basically, Case C  is interesting to show the attack techniques have to generate an adversarial image from the transformed image (after the pre-processing method) in order to fool the neural network effectively.
> > > > > > > In other words, this scenario is equivalent to the neural network has no any defense mechanism.
> > > > > > >
> > > > > > > Summary:
> > > > > > > For Case A, attackers generate A(F, x), neural network uses PR(.) to defend.
> > > > > > > For Case B, attackers generate A(F, x), neural network uses no defense mechanism.
> > > > > > > For Case C, attackers generate A(F, PR(x)), neural network uses no defense mechanism.
> > > > > > > If there is Case D (F(PR(A(F, PR, x)))), attackers generate A(F, PR(x)), neural networks uses PR(.) to defend.
> > > > > > >
> > > > > > > This Case C is also very interesting because Case C can be considered as an insider attack or insider threat which is very crucial in cyber-security.
> > > > > > >
> > > > > > > The attacker can intrude the system and perform deception attack after the image is pre-processed.
> > > > > > > Actually, in view of computer security, the scenario like the case C is very important.
> > > > > > > The attacker can intrude the system, place himself in the middle of the "pre-processing --- classification" pipeline, try to perform deception attack.

---

> > > > > > > > ### Public Comment · (anonymous) · 2018-10-29
> > > > > > > > **Case C: Unreasonable threat model + incorrect evaluation**
> > > > > > > >
> > > > > > > > Now that I believe I completely understand what you are saying, I will make one final comment here.
> > > > > > > >
> > > > > > > > The threat model you study in Case C has no basis in reality. On at least three levels, the evaluation of this case is fundamentally flawed.
> > > > > > > >
> > > > > > > > To begin, this case has nothing to do with insider threats, yes, insider attacks are definitely important. However, insider threats have *nothing* to do with this Case C threat model. If the attacker could somehow perturb an input in between the PR and the F function calls, they could almost certainly do much, much worse. As just a short list of things the attacker could do instead:
> > > > > > > > - Modify the PR function to make it an effective no-op but still make changes to the input so it "looks" like it is doing the right thing.
> > > > > > > > - Modify the F function to make it output the incorrect predictions by making subtle modifications to the weights.
> > > > > > > > - Make unrestricted changes to the input x before it is passed to the neural network F.
> > > > > > > >
> > > > > > > > But fine, let's assume for some reason the only thing the adversary is allowed to do is modify the image PR(x) and they have to make a small perturbation. Even still, the way that you qualitatively measure this attack is wrong. Of course "perturbed pixels from the adversarial image can be perceptible by the human." when looking at A(F,PR(x)) Why? Because PR(x) introduces a very large distortion to begin with! And so even the attacker was just the function F(x) = x, it still appear to have a high distortion.
> > > > > > > >
> > > > > > > > Okay, so even with that out of the way, the attack doesn't even seem to be implemented correctly. The claimed model accuracy at eps=0.3 is ~40% on MNIST, on the images that have been already modified by PR. Recall that in this case, as you say, there is nothing standing in the way of the attacker. The result of the attacker will be fed *directly* into the classification neural network. Because the un-protected neural network has accuracy 0% at eps=0.3, the attacker in Case C should be able to achieve this same result. (Worse yet: it should be even *easier*. They receive as input images that are already noisy.)

---

> > > > > > > > > ### Author Response · Authors · 2018-10-30
> > > > > > > > > **Reply to "Case C: Unreasonable threat model + incorrect evaluation"**
> > > > > > > > >
> > > > > > > > > Thank you for your comments.
> > > > > > > > >
> > > > > > > > > We appreciate your comments and we will take your opinions into account in the future.
> > > > > > > > >
> > > > > > > > > However, we have to clarify that we are not solving the insider threats problem in Case C. We mentioned the insider threats only for a scenario example to the readers, so they could have better understanding.
> > > > > > > > >
> > > > > > > > > Furthermore, we agree that the attacker will receive a very high distortion for generating an adversarial image in Case C due to the PR method.  If we compute |PR(x) - A(F, PR(x))|, the distortion is low and the distortion is high when we compute |x - A(F, PR(x))|. Due to the limitation of the state-of-the-art attacks that we have known, which the attackers have weak ability to deal with the pre-processing method, we show Case C as it is. As aforementioned, the purpose of studying Case C is just to show the attackers have to generate A(F, PR(x)) in order to fool the neural network effectively until we find a proper attack for it. This case also could raise the awareness of the attackers to deal something with the pre-processing method but not only with the neural network itself.
> > > > > > > > >
> > > > > > > > > We claimed that our model can achieve accuracy ~40% at eps=0.3 is because we have trained our model with distorted images (not with clean images) which increased the robustness of the model. This performance can be claimed from the case when the model has trained with the adversarial training which the trained model can maintain a very high performance even the trained model is attacked by the same attack technique (or weaker than the used attack technique) that has used during the adversarial training.
> > > > > > > > > As we have mentioned in the paper, our method is similar to the adversarial training. But, the difference is that the adversarial training uses adversarial images as the training images, whereas our method uses distorted images (may have some chances to generate adversarial images) as the training images. In addition, our method will pre-process the image. Readers can think as our PR method has double defense mechanisms.

---

### Public Comment · (anonymous) · 2018-10-25
**How do you limit CW distortion?**

Byu default, CW will minimize the distortion subject to the fact the image is adversarial. However, it looks like (in Table 2-4) you limit the CW distortion to .3 on MNIST and 0.03 on CIFAR. How do you do this? Also, CW is a L_2 based metric by default. Are you using the L_infinity method instead?

---

> ### Author Response · Authors · 2018-10-26
> **Reply to "How do you limit CW distortion?"**
>
> Thank you for your comments.
>
> To remove the confusion, we would like to rephrase the sentence from
> "During the experiments, we set epsilon = 0.3 for MNIST dataset and 0.03 for Fashion MNIST and CIFAR-10 datasets when we apply with FGSM, BIM, and MIM attacks."
> to
> "During the experiments, when we apply with FGSM, BIM, and MIM attacks, we set epsilon = 0.3 for MNIST dataset and 0.03 for Fashion MNIST and CIFAR-10 datasets."  in the next version.
> For the conclusion, we do not set the epsilon value in CW attack.
>
> We only test CW with the L_2 based metric in this version.
> Thanks to your comment, we would like to add the L_infinity test in our paper next version.

---

> > ### Public Comment · (anonymous) · 2018-10-26
> > **Adaptive Attacks?**
> >
> > Have you tried an attack that uses information about the defense?

---

> > > ### Author Response · Authors · 2018-10-26
> > > **Could you please elaborate the question?**
> > >
> > > We apologies that we do not get your question fully.
> > > Could you please to elaborate your question in details?

---

> > > > ### Public Comment · (anonymous) · 2018-10-26
> > > > **Clarifying the question**
> > > >
> > > > I am not the GP comment, but what they are asking is if you attempted to actively evade your defense by building an attacker who is aware of what the defender is doing. Given your response below, it looks like the answer is "no", you do not assume the attacker ever has access to the PR function. Is this right?

---

> > > > > ### Author Response · Authors · 2018-10-27
> > > > > **Reply to "Adaptive Attacks?" and "Clarifying the question"**
> > > > >
> > > > > Thank you for the comments.
> > > > > As we have replied in "Reply to 'Why is this interesting?'" below, Case C is the case that the attackers have the knowledge of our defense technique. However, our defense technique increases the distortion of the adversarial image. Hence, we mentioned that Case C is impractical.
> > > > >
> > > > > For the conclusion, we did test the attack techniques with the parameters of our defense technique (i.e. Case C).

---

> > ### Public Comment · (anonymous) · 2018-10-26
> > **For unbounded attacks, report distortion; not accuracy percentage**
> >
> > Unbounded attacks (like CW) should succeed ~100% of the time. With an unbounded perturbation, you should always be able to cause a classifier to return the wrong output. You should instead be reporting the perturbation size.
> >
> > That is, it is not helpful to report "CW success rate is 82%". Reporting "CW success rate is 82% with a mean successful perturbation of 3.2." is at least useful, but probably indicates the attack is somehow broken. Best is to report "Mean perturbation is 3.5 to generate an adversarial example."

---

> > > ### Author Response · Authors · 2018-10-27
> > > **Reply to "For unbounded attacks, report distortion; not accuracy percentage"**
> > >
> > > Thank you for the comments.
> > > We will report the distortion value in the next version.

---

### Comment · AnonReviewer2 · 2018-10-29
**some clarifications**

I have some questions that needs to be clarified:
1) In Fig 3, you use perceptron to predict the value of the pixel, how to ensure that the outcome of the perceptron is within the normal range? What would be the case if the input is on the boundary (say close to .5 in graysclae case) then the outcome could be lower or higher than .5?
2) "During the experiments, we set ε = 0.3 for the MNIST dataset and ε = 8/256 for the Fashion MNIST and CIFAR-10 datasets when we apply with the FGSM, BIM, and MIM attacks" How do you justify for these hyper-parameter choices?
3) In section 3.4, you explained multiple cases for different attack mechanisms. These cases for testing the CNN not for training them, right? In other words, do you assume that in the training phase the PR method was used? If this is the case, what is the benefit of using PR in the testing phase rather than attack, i.e. how could using PR in the testing be useful?
4) In Figure 4, the adversarial images in both cases A and B should be the same, right? Why are they different? Unless by adversarial image you sometimes mean the outcome of the PR model as well.
5) In Table 2, why is the accuracy performance on training phase less than the performance on both phases? I would expect that both phases is more challenges because in the test you introduce images with some additional variations.

---

> ### Author Response · Authors · 2018-10-30
> **Reply to "some clarifications"**
>
> Thank you for your comments.
>
> 1. The outcome of the perceptron mostly will be within the normal range if the perceptron is converged. Since the perceptron will give only 1 output, then the outcome to change the pixel has to be based on the output of the perceptron.
>
> 2. We usually follow the default parameters on the scripts or we use the parameters that have been suggested from most related papers.
>
> 3. Yes, the cases in Section 3.4 are only used in the testing phase, after we have trained our model with PR method. In other words, we use PR method in all cases during the training phase except for Section 4.1 and 4.4. By using PR method, it can destroy adversarial pixels with some chances.
>
> 4. We would like to apologies with the caption of the table that we have used in current version and we would like to modify it in the next version to prevent the confusion. In Figure 4, we show PR(A(F,x)) images for Case A and A(F,x) images for Case B.
>
> 5. We apologies that we cannot get this question fully. In Table 2, the accuracy performance on the training phase is higher than the performance in the both phases, mostly.

---

### Comment · AnonReviewer3 · 2018-11-01
**Reply to: "Reply to "Questions on how the technique works""**

Thanks for your response.  The algorithm is much more clear to me now.  I have one more follow up question:
	- What use does the perceptron provide? Why does the PR model need to be trained at all,  given that you can easily exactly the compute the underlying sampling process?  i.e. why not just discretize into one of the color bins (e.g. map 18 to class 1, since you know the range), and then randomly sample from the known range for class 1 to generate the output?

---

> ### Author Response · Authors · 2018-11-01
> **Reply to "Reply to: "Reply to "Questions on how the technique works"""**
>
> Thank you for your comments.
>
> It is true that we can do the sampling directly as stated in your comments if the perceptron is fully converged and only one perceptron is used. However, as we have mentioned in Section 2.2, we generate multiple PR models. The PR models includes partially converged models and fully converged models. If we use the partially converged model to predict a pixel, we can predict the pixel into a different class from its original class. For example, a pixel origin from the class 1 but the model predicts it to class 3. The pixel will no longer be a black color pixel but it turns to a white color pixel. In summary, we use PR model to predict a pixel not only with the same class, but also with a different class.

---

### Meta-Review · Area_Chair1 · 2018-12-16
**Majority reject.**

**Confidence:** 3
**Recommendation:** Reject

**Metareview:**

Based on the majority of reviewers with reject (ratings: 4,6,3), the current version of paper is proposed as reject.